# LOG: Active Model Adaptation for Label-Efficient OOD Generalization

**Jie-Jing Shao, Lan-Zhe Guo, Xiao-Wen Yang, Yu-Feng Li**[*]
National Key Laboratory for Novel Software Technology
Nanjing University, Nanjing 210023, China
{shaojj, guolz, yangxw, liyf}@lamda.nju.edu.cn

## Abstract

This work discusses how to achieve worst-case Out-Of-Distribution (OOD) generalization for a variety of distributions based on a relatively small labeling cost. The problem has broad applications, especially in non-i.i.d. open-world scenarios. Previous studies either rely on a large amount of labeling cost or lack of guarantees about the worst-case generalization. In this work, we show for the first time that active model adaptation could achieve both good performance and robustness based on the invariant risk minimization principle. We propose LOG, an interactive model adaptation framework, with two sub-modules: active sample selection and causal invariant learning. Specifically, we formulate the active selection as a mixture distribution separation problem and present an unbiased estimator, which could find the samples that violate the current invariant relationship, with a provable guarantee. The theoretical analysis supports that both sub-modules contribute to generalization. A large number of experimental results confirm the promising performance of the new algorithm.

## 1 Introduction

Machine learning models are typically trained and tested on the same data distribution. However, when these models are deployed in real task scenarios, they face much inapplicability, because the data distribution of the target task usually deviates from that of training. For example, the financial data prediction model based on local users is often inaccurate in predicting user behavior in other regions. Similar examples include self-driving [42], speech recognition [16], influenza detection [30], etc.

To this problem, the recent research on *OOD (Out-Of-Distribution) Generalization* [23, 7, 1, 13, 40, 19, 5] has given a series of technologies, trying to obtain models with robustness, which have certain worst-case generalization guarantees on a variety of unseen distributions. Although they do not access target data, they need a large number of high-quality source data to remove the source-specific spurious correlation and obtain the generalizable invariant relationship, i.e., sufficient multi-source labeled data and accurate source information. However, these two requirements are still difficult to satisfy in most practical tasks.

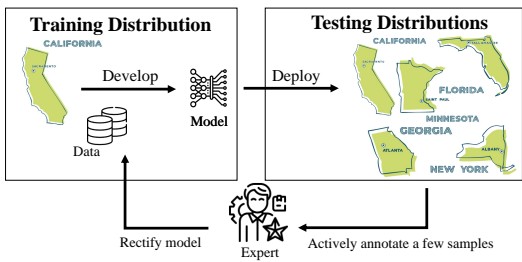

Figure 1: Example of model adaptation to a variety of distributions. We would like to annotate a few samples from target testing distributions to rectify the model in the source iteratively.

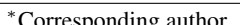

[*]Corresponding author

36th Conference on Neural Information Processing Systems (NeurIPS 2022).

There are also some more traditional approaches to handle distribution shift, such as *Domain Adaptation* [25, 9, 20, 21]. They introduce unlabeled data from the target distribution, reducing the requirements on the source data relative to OOD generalization. However, they typically assume that target samples are from an isolated homogeneous distribution, which ignores the generalization in varied distributions. Recently, there are some works [10, 26, 22, 33] focusing on the adaptation to multiple latent domains. Nevertheless, the robustness, i.e., worst-case guarantees has not been addressed, especially without explicit domain labels.

To make a reasonable compromise between strongly labeled information dependence and robustness, *Active Model Adaptation* may be a feasible scheme. It gradually increases the labeling cost and actively annotates the data in the target task that is difficult to be robustly generalized. Such a scheme could effectively reduce the strong dependence on label information. It has shown promising performance when the target distribution is an isolated homogeneous distribution [39, 28, 6, 8].

In this paper, we would like to derive a benefit from *Active Model Adaptation* to address robustness at a small labeling cost. We first propose the invariant risk minimization principle for active model adaptation. That is, we could tend to both overall performance and worst-case guarantee via maximal invariant predictor. Based on this, we further present an interactive framework to achieve label-efficient OOD generalization (LOG), composed of an actively querying module and an invariant learning module. The key challenge is to find unlabeled samples where the current invariant model does not hold. Based on the structural causal model assumption, we formulate it as a mixture distribution separation problem and present an unbiased estimator to address it. In theory, our actively querying module could accurately find the samples which violate the current invariant relationship. In the experiments on a series of tasks, the promising performance of our LOG has been confirmed.

## 2 Related Work

**OOD (Out-of-Distribution) Generalization** works on learning models that generalize well on a variety of unseen distributions. There are two main branches: *Domain Generalization* [23, 7, 40] mostly focuses on computer vision problems as predictions are prone to be affected by a disturbance on images (e.g., style, background, etc). *Causal & Invariant Learning* starts from causal inference and explores causal variables to address generalization ability under covariate shift [1, 13, 35]. Most of them rely on labeled data from multiple sources to remove the source-specific spurious correlation. Recently, some works [19, 5] attempt to mine the heterogeneity of an assembled source without explicit prior division. Nevertheless, they commonly assume source data is sufficient to learn the invariance and ignore the exploration of data from target distributions.

**Domain Adaptation** works on addressing the domain shift between the training source and the testing target, where no labeled data are available in the target domain [25, 9, 20, 21]. They typically assume the target samples are from a single homogeneous domain. The most related subtopic to us is *Latent Domain & Domain-Agnostic Adaptation* that focuses on the target distribution with multiple latent domains [26, 22, 33]. Although they propose methods with good overall performance in the presence of multiple latent distributions, the worst-case guarantee has not been addressed.

**Active Model Adaptation** works on active learning under distribution shift. Previous works have focused on two types of distribution shifts. 1) label space shift: adapt a pre-trained model to a task with different label spaces [11]; 2) domain shift: adapt a source model to a target domain [39, 28, 6, 8, 34]. To the best of our knowledge, all of them directly regard the target distribution as a homogeneous domain and ignore the generalization ability when massive distributions exist.

## 3 Problem and Analysis

Generally speaking, the above technologies are difficult to directly deal with the model generalization of a variety of distributions under a small labeling cost. To deal with such a challenge, in this paper, focusing on active model adaptation, we first analyze the theoretical basis of the problem. Based on this, we present a new active adaptation principle, invariant risk minimization. We further put forward the corresponding algorithm, and show its effect on generalization ability.

### 3.1 Problem Formulation

In the source training stage, the learner collects a sufficient labeled dataset $D_S = \{D^e\}_{e \in \mathcal{E}_S}$, which is a mixture of data $D^e \in \mathcal{X} \times \mathcal{Y}$ collected from the collection of training sources $e \in \mathcal{E}_S$. $\mathcal{X}$ and $\mathcal{Y}$ denote an input and output space, respectively. Source model $f_S : \mathcal{X} \rightarrow \mathcal{Y}$ is well-trained on the source dataset $D_S$ with a small risk $R(f_S; D_S)$. When the model is deployed in open-world scenarios, it needs to adapt to the distributions $\mathcal{E}_T$, i.e. a collection of varying testing distributions. Following [26, 33], we formulate the target distribution $\mathcal{D}_T \in \mathcal{X} \times \mathcal{Y}$ as a mixture of base distributions: $\mathcal{D}_T = \sum_{e \in \mathcal{E}_T} \lambda^e \mathcal{D}^e$, where $\mathcal{D}^e$ is the distribution observed from the environment $e$ and $\lambda^e$ represents the corresponding proportion. Note that in reality, data are frequently assembled under implicit environment information $e$ and $\lambda^e$, thus we do not depend on this information to develop algorithms. They are used only during evaluation. To fast adapt the source model to the target distribution, active learning has been introduced [39, 28, 8, 34]. Generally, their goal is:

**Definition 3.1** (Performance Maximization). The performance goal is to minimize the generalization risk of model $f$ on the overall target distribution $\mathcal{D}_T$ based on some queried samples $Q$:

$$\min_{Q, \mathcal{A}} R(\mathcal{A}(f_S; \{D_S, Q\}); \mathcal{D}_T) \tag{1}$$

where $\mathcal{A}$ is a model adaptation algorithm to rectify $f_S$: $f \leftarrow \mathcal{A}(f_S; \{D_S, Q\})$.

In addition to the optimization of ideal overall generalization, we also need to comprehensively consider the robustness, i.e., worst-case generalization on $\mathcal{E}_T$:

**Definition 3.2** (Robustness Preservation). The robustness means we could maintain low risk even in the worst distribution $\mathcal{D}^e$ across $\mathcal{E}_T$, i.e. worst-case guarantee on each base distribution $\mathcal{D}^e$.

$$\min_{Q, \mathcal{A}} \max_{e \in \mathcal{E}_T} R(\mathcal{A}(f_S; \{D_S, Q\}); \mathcal{D}^e) \tag{2}$$

$Q$ and $\mathcal{A}$ are important to address active model adaptation. Previous works [11, 39, 28, 8] mainly focus on the $Q$ to address performance maximization. They do not consider the choice of $\mathcal{A}$ (simply take it as standard supervised learning or semi-supervised domain adaptation) and the robustness.

### 3.2 Invariant Risk Minimization

Without any prior knowledge or structural assumptions, it is impossible to adapt the source model to target distributions, since one cannot characterize the shift between $\mathcal{E}_S$ and $\mathcal{E}_T$. Following the [1, 13, 2, 24, 38, 17], we consider the data generation via a structural causal model under the covariate shift assumption (i.e., the $P(y|x)$ is unchanged across the varying distributions):

**Assumption 3.3.** Consider a structural causal model [41] governing the random distribution $P(X, Y)$ and the learning goal of predicting $Y$ from $X$. Then the set of all distributions $\mathcal{E}$ indexes all the interventional distribution $P^e(X^e, Y^e)$ obtainable by valid interventions $e$:

$$X^e = S(Z_s, Z_v^e), Y^e = f(Z_s) + \epsilon, \epsilon \perp X^e. \tag{3}$$

$Z_s$ and $Z_v^e$ represent the semantic variable and intervention variable, respectively. The feature $X^e$ could be regrad as an observation $S$ for $Z_s$ in the intervention variable $e$, influenced by the $Z_v^e$. The label $Y$ is caused by the semantic $Z^e$ and an independent noise term $\epsilon$. We assume the data generation process $S$ is inevitable and there exists $\Phi(S(X^e)) = Z_s$ to recover the semantics for all $Z_s$ and $Z_v^e$.

**Remark 3.4.** *Following the above assumption about structural causal model, the ideal $\Phi^*$ satisfies: 1) The margirial invariance $P^e(\Phi^*(X)) = P^{e'}(\Phi^*(X))$ and conditional invariance $P^e(Y|\Phi^*(X)) = P^{e'}(Y|\Phi^*(X))$ hold for any $e, e' \in \mathcal{E}$. 2) It is sufficient to predict the target using $\Phi^*(X)$ as the input: $Y = f(\Phi^*(X)) + \epsilon, \epsilon \perp X$, across varying $e \in \mathcal{E}$.*

To acquire such $\Phi^*(X)$, a branch of invariant risk minimization proposals [1, 13, 19, 5] finds the *Invariant Features* and the corresponding *Maximal Invariant Predictor*, defined as:

**Definition 3.5** (Invariant Features). The set of invariant features $\mathcal{I}$ with respect to $\mathcal{E}$ is defined as:

$$\mathcal{I}_\mathcal{E} = \{\Phi(X) : H[Y|\Phi(X)] = H[Y|\Phi(X), \mathcal{E}], \Phi(X) \perp \mathcal{E}\}$$

where $H[\cdot]$ is Shannon entropy of a random variable. The corresponding *Maxiaml Invariant Predictor* (MIP) of $\mathcal{I}_\mathcal{E}$ is defined as: $\Phi^* = \arg\max_{\Phi \in \mathcal{I}_\mathcal{E}} I(Y; \Phi)$ where $I(\cdot; \cdot)$ measures Shannon mutual information between two random variables.

Here we prove that we could achieve both performance and robustness based on the MIP $\Phi^*$.

**Theorem 3.6.** *For a predictor $\Phi^*(X)$ in the collection $\mathcal{E}$, the solution $f^*(X) = P(Y|\Phi^*)$ could achieve both performance maximization $R(f^*; \mathcal{D}_T) \leq \min_f R(f; \mathcal{D}_T)$ and worst-case robustness $\max_{e \in \mathcal{E}} R(f^*; \mathcal{D}^e) \leq \min_f \max_{e \in \mathcal{E}} R(f; \mathcal{D}^e)$.*

Although the ideal MIP has shown a favorable theoretical guarantee, we find its generalization is heavily dependent on the heterogeneity of source distributions collection $\mathcal{E}_S$. We further analyze the connection between generalization and source heterogeneity and propose an active model adaptation framework to expand the heterogeneity and trend to the ideal MIP.

### 3.3 Active Heterogeneity Expansion

Given source collection $\mathcal{E}_S$ and the target collection $\mathcal{E}_T$, denote the corresponding invariant features set $\mathcal{I}_S$ and $\mathcal{I}_T$ respectively. For $\mathcal{E}_S \subseteq \mathcal{E}_T$, the corresponding invariant features set satisfies $\mathcal{I}_T \subseteq \mathcal{I}_S$ [19]. It indicates the generalization of $\mathcal{I}_S$ could be improved by extending the source $\mathcal{E}_S$ and promoting the $\mathcal{I}_S$ to ideal $\mathcal{I}_T$. Here, we present the generalization condition for $\Phi_S$ to an unseen distribution $e'$ and then justify that the generalization of $\Phi_S$ is heavily dependent on the heterogeneity of $\mathcal{E}_S$.

**Theorem 3.7.** *For distribution $P^{e'}(X^{e'}, Y^{e'})$, if $\Phi(X^{e'}) = \Phi(X^e)_{e \in \mathcal{E}_S}$, $\mathcal{I}_S$ is equal to the invariance set constrained by $\mathcal{E}_S \cup \{e'\}$. The optimal source model $f_S$ could generalize on the distribution $P^{e'}$. The generaliable distributions: $\mathcal{E}_G = \{e'|\mathcal{I}_S = \mathcal{I}_{\mathcal{E}_S \cup \{e'\}}\} = \{e'|\Phi(X^{e'}) = \Phi(X^e), e \in \mathcal{E}_S\}$.*

*Particularly in the linear structural causal model, we further assume the semantics $Z_s$ takes values in $\mathbb{R}^c$, intervention $Z_v^e$ takes values in $\mathbb{R}^w$, and $S \in \mathbb{R}^{d \times (c+w)}$. Let $\Phi \in R^{d \times d}$, we have:*

$$\mathcal{E}_G = \{e'|\mathbb{E}[X^{e'}] = \mathbb{E}[X^e] - x, e \in \mathcal{E}_S, x \in \ker(\Phi)\} \tag{4}$$

*where $\dim(\ker(\Phi)) = \dim(\text{span}(\{\mathbb{E}[X^e]\}_{e \in \mathcal{E}_S})) - 1$.*

**Remark 3.8.** *Theorem 3.7 propose the generalization condition for any unseen $e'$. Compared with previous analysis [1, 27] in linear cases, the Equation 4 further indicates a quantitative dependence of generalization on source heterogeneity. In words, the freedom of $\mathcal{E}_G$ and the nullspace $\ker(\Phi)$ is limited by the maximal linearly independent system (heterogeneity) from source collection $\mathcal{E}_S$.*

Theorem 3.7 motives us to expand collection $\mathcal{E}_S$ via querying samples from a distribution $P^{e'}$ where $\Phi(X^{e'}) \neq \Phi(X^e)_{e \in \mathcal{E}_S}$. Formally, we have the following proposition:

**Proposition 3.9** (Active Heterogeneity Expansion)**.** *To address performance maximization and robustness, we could guide the active model adaptation via the heterogeneity expansion:*

$$\mathcal{A}(f_S; Q) \to f^* = P(Y|\Phi_T^*), \tag{5}$$

*promoting the $\Phi_S$ of $f_S$ to ideal $\Phi_T$. At each active data collection stage, we query and collect the samples from $\mathcal{E}_T \backslash \mathcal{E}_G$ to expand the data heterogeneity.*

**Theorem 3.10.** *Under the linear structural causal model, when we collect a distribution $e'$ where $\Phi(X^{e'}) \neq \Phi(X^e)_{e \in \mathcal{E}_S}$, and update $\Phi \to \Phi'$, $\text{rank}(\Phi') = \text{rank}(\Phi) - 1$ holds. The $\Phi$ will converge to the ideal $\Phi^*$ at most $w$ iterations, where $w$ is the freedom of the intervention variable $Z_v^e \in \mathbb{R}^w$.*

Theorem 3.10 indicates our proposed active heterogeneity expansion framework shares a favorable convergence rate. In words, each time we collect a distribution from $\mathcal{E}_T \backslash \mathcal{E}_G$, we can remove one degree of freedom in the space of the variant intervention factor. It is noteworthy that although the convergence analysis is based on the linear condition, our proposed framework is still applicable in non-linear cases where the condition $\mathcal{E}_G = \{e'|\Phi(X^{e'}) = \Phi(X^e), e \in \mathcal{E}_S\}$ in Theorem 3.7 holds.

## 4 Algorithm

Following the above analysis, we would like to query the samples from $\mathcal{E}_T \backslash \mathcal{E}_G$ to achieve the ideal $\mathcal{I}_T$. In this work, we propose our active model adaptation method LOG, with two interactive modules:

1) *Query Strategy $\mathcal{M}_Q$*: given the unlabeled data pool $X_T$ and the current invariant relationship $\mathcal{I}$, actively query the representative samples $Q$ from the un-generalizable distributions $\mathcal{E}_U = \mathcal{E}_T \backslash \mathcal{E}_G$.

2) *Model Adaptation $\mathcal{M}_A$*: given the newly queried samples $Q$, update the invariant relationship $\mathcal{I}$ via invariant learning.

The whole framework is iterative so that the mutual promotion between active exploration and invariance exploitation can be leveraged.

## 4.1 Query Strategy Module $\mathcal{M}_Q$

Following the problem formulation, we have the $X_T = \theta X_G + (1-\theta)X_U$, where $X_G = P(X|\mathcal{E}_G)$, $X_T = P(X|\mathcal{E}_T)$. The $X_U$ reperesents the obversation from the un-generalizable $\mathcal{E}_U = \mathcal{E}_T \backslash \mathcal{E}_G$.

Based on Proposition 3.9, we can query few instances $Q$ from $X_U$ to promote $\Phi_{\{D_S, Q\}} \to \Phi_T^*$. This goal could be formulated as:

$$\mathcal{M}_A([X_s, y_s], [X_Q, y_Q]) \to \mathcal{M}_A([X_s, y_s], [X_U, y_U])$$

We first detect the samples from $X_U$ and then find the representative instances $X_Q$ for $X_U$ to query. Based on the connection $\Phi(X_S) = \Phi(X_G)$, we could transform the detection of $X_U$ as a mixture distribution separation problem. We denote $\mathbb{I}(x_i \sim X_G)$ to indicate if $x$ is observed from the $\mathcal{E}_G$. The indicator is essentially a binary classifier $g \circ \Phi(x) : \mathcal{X} \to \{1, -1\}$. Given a sample $x$, when $g \circ \Phi(x) = 1$, it is observed from $\mathcal{E}_G$, and the current model could give a correct prediction, otherwise it will violate the current invariance $\Phi$ and is risky for the current model.

One of the main challenges to learning $g$ is that we do not have information about the marginal distribution of unseen $\mathcal{E}_U$. We handle this problem by using the risk rewriting technique [12, 43] with the unlabeled data from $\mathcal{E}_T$.

$$P(\Phi(X_U)) = (P(\Phi(X_T)) - \theta P(\Phi(X_G)))/(1-\theta) \tag{6}$$

**Proposition 4.1.** *For all measurable function g, we could rewrite the original risk as:*

$$\begin{aligned} R(g) &= \theta E_{x \sim X_G}\left[\ell(g \circ \Phi(x), 1)\right] + (1-\theta)E_{x \sim X_U}\left[\ell(g \circ \Phi(x), -1)\right] \\ &= \theta E_{x \sim X_G}\left[\ell(g \circ \Phi(x), 1) - \ell(g \circ \Phi(x), -1)\right] + E_{x \sim X_T}[\ell(g \circ \Phi(x), -1)] \end{aligned}$$

Since the risk equals the ideal risk, its empirical estimator $\hat{R}(g)$ is unbiased over the target distribution. We can thus perform the standard empircal risk minimization. Particularly when the binary loss satisfying $\ell(z) - \ell(-z) = -z$, for all $z \in \mathbb{R}$, the empirical risk $\hat{R}(g)$ is convex w.r.t. $g$ [12] [2]. It leads to a convex optimization problem, for which the globally optimal solution can be obtained.

Here we further analyze the estimation error of indictor $g$ based on [12]. Formally, the empirical estimator $\hat{g}$ has the estimation error bound:

**Theorem 4.2** (Estimation Error Bound). *Suppose that $\inf_{g \in \mathcal{G}} R(g) \geq \alpha > 0$ and $\mathcal{G}$ is closed under negation, i.e. $g \in \mathcal{G}$ iff $-g \in \mathcal{G}$. Then for any $\delta > 0$, with probability at least $1 - \delta$,*

$$R(\hat{g}) - \inf_{g \in \mathcal{G}} R(g) \leq \mathcal{O}_p\left(\frac{1}{\sqrt{N_S}} + \frac{1}{\sqrt{N_T}}\right). \tag{7}$$

*For a better presentation, we use the $\mathcal{O}_p$-notation to keep the dependence on $N_S$ and $N_T$ only.*

Theorem 4.2 shows that the estimation error of the estimated $\hat{g}$ decreases with a growing number of source data $N_S$ and unlabeled target data $N_T$. In our problem, we have the plentiful source data $D_S$ from $\mathcal{E}_S$ and *unlabeled data $X_T$* from $\mathcal{E}_T$, which helps to estimate $\hat{g}$ accurately.

Notice that the implementation of our algorithm requires the knowledge of the mixture proportion $\theta$, where plenty of works [3, 31, 29] have been explored to estimate $\theta$ from the unlabeled data. We adopt the mixture proportion estimation method of [29] in our implementation and omit the details here. All of our empirical studies are conducted by the estimated mixture proportion.

Based on the distribution inference module $g$, we could get the probability $p_u(x) = p(-1|x; g \circ \Phi)$ that $x$ belongs to $\mathcal{E}_U$. Here we further consider to obtain the limited subset $Q$ covering the empirical set $X_U = p_u(x) \cdot X_T$. Specifically, we follow the core-set objective proposed in [32]:

$$\min_Q \max_{x \in X_U} \min_{x' \in Q} \|x - x'\|_2 \leq \delta_Q \tag{8}$$

---

[2]Many popular loss functions satisfy the condition, such as logistic loss, square loss and double hinge loss.

Informally, we are trying to find a subset $Q$ to query labels that are close to the raw candidates $X_U$. Although this problem is NP-hard [4], we could obtain an approximate solution efficiently using the greedy iterative approach:

1) Get $x = \arg\max_{x \in X_T \setminus Q} \min_{x'} p_u(x) \|x - x'\|_2$.
2) Add sample $x$ to the subset $Q$: $Q = Q \cup \{x\}$.

## 4.2 Model Adaptation Module $\mathcal{M}_A$

Given the newly queried samples $Q = \{X_Q, y_Q\}$, we have environments $\mathcal{E} = [(X_S, y_S), (X_Q, y_Q)]$. Then we could update the invariant feature $\Phi$ via standard environment-based invariant learning. Following [14, 36, 15, 19], we mine the invariance on raw feature level, a simple but general setting. We further obtain $\Phi$ through feature selection: $\Phi(X) = M \odot X$.

The objective function of $\mathcal{M}_A$ with $M \in \{0, 1\}^d$ is:

$$\min_{M,f} \sum_{D^e, e \in \mathcal{E}} R(f \circ M | D^e), \quad \text{subject to} \quad f \in \arg\min_{\tilde{f}} R(\tilde{f} \circ M | D^e), \forall e \in \mathcal{E}.$$

However, as the optimization of hard feature selection with binary mask $M$ suffers from high variance, we use the soft feature selection with gates taking continuous value in $[0, 1]$.

LOG is applicable broadly to environment-based invariant learning objectives through the different choices of $\mathcal{M}_A$. In this paper, we choose IGA [13] which has guaranteed to achieve the maximal invariant predictor with respect to given environments.

## 4.3 Interactive Promotion

**Theorem 4.3** (Interactive Promotion). *Given the newly queried samples from $\mathcal{E}_U$, invariant learning module $\mathcal{M}_A$ could promote the invariance set $\mathcal{I}_S$ to ideal $\mathcal{I}_T$. Given the updated invariance set $\mathcal{I}' \subset \mathcal{I}_S$, we have better generalization: $\mathcal{E}'_G \supset \mathcal{E}_G$ and reduced the candidates for actively sampling.*

The core of our LOG framework is the mechanism for $\mathcal{M}_Q$ and $\mathcal{M}_A$ to mutually promote each other. Here we theoretically justify the positive feedback. It indicates that our active exploration could help the reduction of the current invariance set $\mathcal{I}_S$. On the other hand, the better invariance set $\mathcal{I}$ could help to expand the generalizable distributions and reduce the querying candidates.

# 5 Empirical Study

In this section, we provide extensive results to evaluate LOG and compared methods for both benchmark simulation and a series of real-world tasks.

**Competing Methods**  We firstly consider learning only on source: including ERM baseline, and 2 state-of-the-art OOD generalization methods: HRM [19] and EIIL [5], which mine the latent heterogeneity without prior division label $e \in \mathcal{E}_S$. Then we consider active model adaptation, including randomly querying and CoreSet [32] baselines, 3 state-of-the-art active model adaptation methods: AADA [39], CLUE [28], DBAL [6].

We use Logistic Regression and Linear Regression as the base models for classification and regression, respectively. Our feature selection weights $M$ could be derived directly from the parameters of them.

## 5.1 Simulation Data

Synthetic data are important tools to simulate explainable and controllable distributional shifts. As indicated by [2, 37], it is necessary to introduce such simple but challenging data, which can reflect whether and to what extent an algorithm can resist certain kinds of distributional shifts. Generally speaking, the covariates $X$ are divided into $X = [S, V]^T$, corresponding to the invariant and variant parts inside the data. The $P(Y|S)$ remains invariant across distributions. The $P(Y|V)$ is perturbed with different mechanisms, which brings a distribution shift.

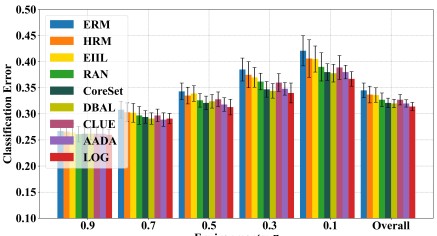
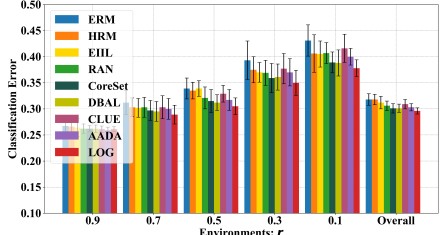

(a) Error under Anti-Causal Effect Shifts

(b) Error under Anti-Causal Effect Shifts (Imbalanced)

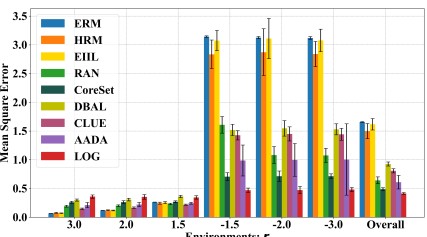
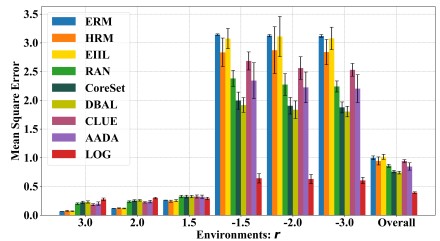

(c) Error under Selection Bias Shifts

(d) Error under Selection Bias Shifts (Imbalanced)

Figure 2: Results on varying base distributions (under 10% labeling budgets).

**Classification with Anti-Causal Effect** Arjovsky et al.(2019) introduce anti-causal relationship to change $P(Y|V)$. Specifically, each distribution is characterized by its bias rate $r \in (0, 1]$, measureing the strength and direction of the spurious correlation between $Y \in \{1, 0\}$ and $A \in \{1, 0\}$. To be detailed, bias rate $r$ represents that for $100 * r\%$ data, $A = Y$, and for the other $100 * (1 - r)\%$ data, $A = 1 - Y$. Then invariant $S$ and variant $V$ are generated as:

$$S|Y \sim \mathcal{N}(Y\mathbf{1}, \sigma_s^2), V|A \sim \mathcal{N}(A\mathbf{1}, \sigma_v^2) \tag{9}$$

We generate 1000 samples from $r_s = 0.9$ as source data $D_S$, 5000 samples from 5 uniform environments $\mathcal{E}_T$ with $r \in [0.9, 0.7, 0.5, 0.3, 0.1]$ as unlabeled data pool $X_T$ (1000 samples for each $r$). We carry out the procedure 10 times and report the average results in Figure 2(a).

**Regression with Selection Bias** Kuang et al.(2020) propose a selection bias mechanism to introduce distributional shifts, and similar settings are also adopted in [19, 18] The data are generated as: $Y = f(S) + \epsilon$ where $\epsilon \perp V$. The selection probability of certain data point $(x, y)$

$$P(x, y) = \prod_{v_i \in V} |r|^{-5*|y - \text{sign}(r)*v_i|} \tag{10}$$

where $|r| > 1$. Intuitively, $r$ controls the strengths and direction of the spurious correlation between $V$ and $Y$. The larger $|r|$ means the stronger spurious correlation between $V$ and $Y$. $r > 0$ means positive correlation and vice versa. Therefore, we can adopt different $r$ to simulate varying distributions.

We generate 2000 samples from $r_s = 2.0$ as source data $D_S$, 3000 samples from 6 uniform environments with $r \in \mathcal{E}_T = [3.0, 2.0, 1.5, -1.5, -2.0, -3.0]$ as unlabeled data pool $X_T$ (500 samples for each $r$). We carry out the procedure 10 times and report the average results in Figure 2(c).

**Imbalanced Mixture** In the real world, there is a natural phenomenon that empirical data follow a power-law distribution. Here we further simulate an imbalanced situation where the source distribution dominates the target distribution collection $\mathcal{E}_T$. Specifically, we generate half of the target samples from $r_S$ and generate the other from different $r$. In this case, it is more difficult to query the target-specific samples than in the uniform case. We report the results in Figure 2(b) and 2(d).

From the results, HRM and EIIL could improve worst-case robustness over the ERM baseline. Without sufficient multi-source labeled data, their robustness is still limited because source heterogeneity

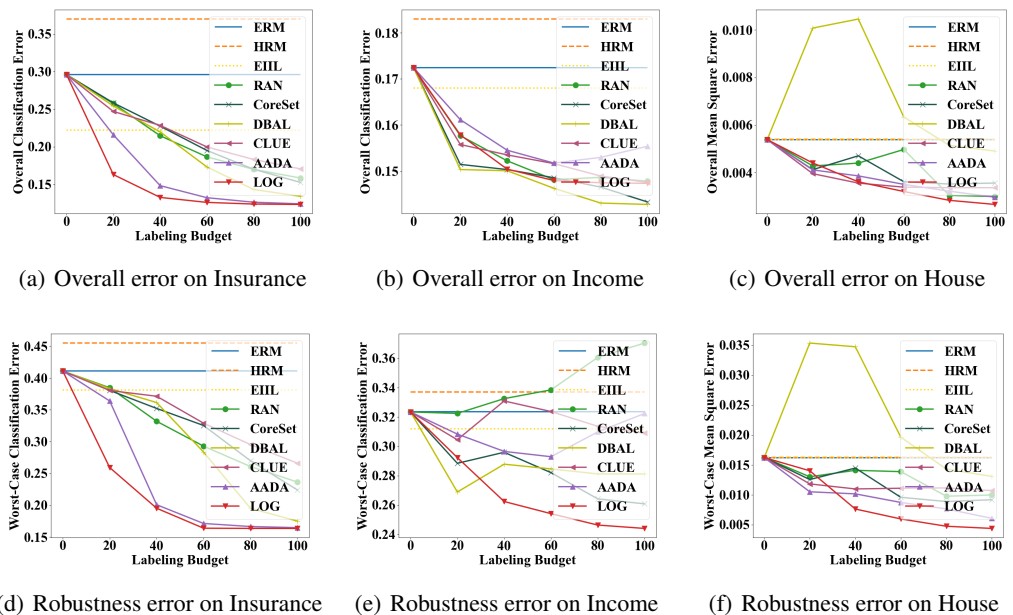

(a) Overall error on Insurance      (b) Overall error on Income      (c) Overall error on House

(d) Robustness error on Insurance    (e) Robustness error on Income    (f) Robustness error on House

Figure 4: Results on real-world tasks. The average results for 10 times produces are reported.

is not sufficient to support varied unseen distributions. In contrast, active adaptation methods which introduce the few labeled data have achieved more significant improvement.

Under the anti-causal shift, all of these active adaptation methods have improved the overall performance of target distributions $\mathcal{E}_T$. Nevertheless, they show significant performance degradation in the base distribution compared to the source distribution. Our LOG has an error which is close to the source error, showing superiority in robustness.

Under selection bias shift, we note that although these adaptation methods have achieved improvement on overall performance, their performance on base distributions (3.0, 2.0, 1.5) close to the source is weaker than source-only methods. To obtain a more clear explanation, we further visualize the dependence of them on invariant variables (S0-S6) and variant variables (V0-V2). One plausible reason is that the source-only methods directly fit the source-specific correlation, overfitting the source distribution. As illustrated in Figure 3, previous methods over-focus on source-specific correlation V2 leads to performance degradation in shifted distributions. In contrast, our method pays more attention to invariant variables (S0-S6) and excludes the variant variables (V0-V2).

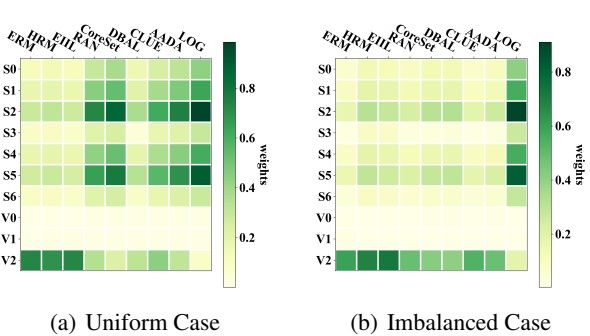

(a) Uniform Case      (b) Imbalanced Case

Figure 3: Feature importance for each method.

## 5.2 Real-world Data Sets

We further evaluate our method on three real-world tasks, including car insurance prediction, people income prediction, and house price prediction, with diverse shift types: region, person, and time.

To evaluate both the overall performance and worst-case robustness, we employ the overall error: $\text{error}(f; D_T)$ and worst-case error: $\max_{e \in \mathcal{E}_T} \text{error}(f; D^e)$.

**Car Insurance Prediction**    In this task, we use a real-world dataset for car insurance prediction (Kaggle). It is a classification task to predict whether 381109 persons will buy car insurance based on related information, such as vehicle damage, and vehicle age[3]. We split the dataset into 7 sub-distributions, according to the *region* of these persons. 50% samples from the first region are split in the source labeled data $D_S$, and the rest is regarded as unlabeled data pool. We report the results under varying labeling budgets in Figure 4(a) and 4(d).

**People Income Prediction**    In this task, we use the Adult dataset to predict personal income levels as above or below $50000 per year based on personal details. There are 48842 instances in this dataset. we split them into 10 groups according to demographic attributes *sex* and *race*. 50% samples from the first group are split in the source labeled data $D_S$, and the rest is regarded as unlabeled data pool. We report the results under varying labeling budgets in Figure 4(b) and 4(e).

**House Price Prediction**    In this task, we use a real-world regression dataset of house sales prices from King County, USA[4]. The target variable is the transaction price of the house. Each sample contains 17 predictive variables, such as the built year, number of bedrooms, square footage of the home, etc. Since it is fairly reasonable to assume the relationships between predictive variables and the target vary along the time (for example, the pricing model may change along the time), there exist distributional shifts in the price prediction task concerning the build year of houses. Here, 50% samples of houses built in [1900, 1990] are split in the source labeled data $D_S$, and the rest samples are regarded as unlabeled data pools. We evaluate model on base distributions through time intervals: [1991, 1995], [1996, 2000], [2001, 2005] and [2006, 2020], to obtain the observations under time shift. We report the results under varying labeling budgets in Figure 4(c) and 4(f).

**Analysis**    From the results, we have the following observations and analyses: The unstable performance of HRM and EIIL indicates the difficulty of generalizing to varied target distributions only by the heterogeneous nature of the source. Randomly querying could perform better than source-only methods at a small labeling cost. In the income task, the source person group dominates the whole distribution, being an imbalanced case. Randomly querying more samples increases the risk of robustness and shows poor worst-case errors. Nevertheless, our LOG has shown stable robustness. CoreSet and CLUE achieve performance gain under distribution shift but are still not satisfying. An interesting phenomenon is that in the early stage of adaptation, DBAL achieved a high improvement in the income task but a significant drop in the house task. One plausible reason is that these samples queried by discrepancy measures are much different from the source data. While they bring rich heterogeneous information, they also increase the risk of the model adaptation part. In contrast, our LOG consistently achieves improvement in overall performance and worst-case robustness, showing the effectiveness of our framework.

### 5.3    Ablation Study

To evaluate the mutual promotion between two sub-modules $\mathcal{M}_A$ and $\mathcal{M}_Q$, we further make the ablation study here. Specifically, we remove $\mathcal{M}_A$ and $\mathcal{M}_Q$ respectively for comparative experiments. In Table 1, we report the overall error and worst-case error on these real-world tasks. It demonstrates that we could tend to both overall performance and robustness through our interaction between active exploration and invariance exploitation.

Table 1: Overall and worst-case error under 100 labels.

|    | $\mathcal{M}_A$ | $\mathcal{M}_Q$ | Insurance | Income | House |
|----|----|----|----|----|----|
|    | $\checkmark$ |    | .131±.008 | .154±.003 | .296±.029 |
| O. |    | $\checkmark$ | .128±.005 | .153±.014 | .344±.205 |
|    | $\checkmark$ | $\checkmark$ | **.123±.000** | **.150±.002** | **.266±.012** |
|    |    | $\checkmark$ | .185±.028 | .277±.027 | .949±.190 |
| W. |    | $\checkmark$ | .168±.005 | .368±.139 | .578±.391 |
|    | $\checkmark$ | $\checkmark$ | **.164±.000** | **.235±.006** | **.439±.052** |

[3]https://www.kaggle.com/anmolkumar/health-insurance-cross-sell-prediction
[4]https://www.kaggle.com/c/house-prices-advanced-regression-techniques/data

## 6  Conclusion

In this paper, we study the active model adaptation to rectify a known model adapting to a variety of distributions. To our best knowledge, we first introduce the invariant risk minimization principle to guide active adaptation, which leads us to optimal performance and worst-case robustness. Based on the structural causal model assumption, we find the generalization could be significantly improved by expanding the heterogeneity of training data. It motivated us to actively expand the data heterogeneity. We further propose an algorithm LOG that integrates query strategy and invariant model adaptation, with an unbiased estimator to detect the un-generalizable samples. We theoretically justify the mutual promotion relationship between our two sub-modules, resonating with the joint process. A series of empirical studies validate the effectiveness of our algorithm in terms of performance and robustness.

This work focuses on the active model adaptation and provides a promising perspective on label-efficient OOD generalization. Our framework mainly focuses on the raw-level feature selection and the corresponding empirical studies are conducted on tabular tasks. The current proposal is not applicable to the high-dimensional data modality, such as image data. We will put efforts to integrate the power of representation learning capabilities of neural networks and explore broader applications.

## Acknowledgement

This research was supported by the National Science Foundation of China (62176118, 61921006), and the Nanjing University-Huawei Joint Research Program. We are grateful to the anonymous reviewers for their helpful comments.

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
