# OpenReview forum: "LOG: Active Model Adaptation for Label-Efficient OOD Generalization"
_NeurIPS.cc/2022/Conference — NeurIPS 2022 Accept_

### Official Review · Reviewer_nvJG · 2022-07-06

**Rating:** 8
**Confidence:** 4
**Soundness:** 4 excellent
**Presentation:** 3 good
**Contribution:** 3 good

**Summary:**

This paper considers Out-of-Distribution generalization via actively querying samples at a relatively small labeling cost. The authors propose an invariant minimization principle and the corresponding active heterogeneity expansion to implement it in an interactive framework. The proposed RA2 framework is theoretically sound and well-validated in sufficient experiments. More specifically, the authors present the generalization condition and a quantitative dependence on source heterogeneity under the linear causal structure. The convergence of RA2 is provided under the linear assumption. The experiments include both benchmark simulation and a series of real-world tasks, supporting the effectiveness of RA2.

**Questions:**

1.	Could the whole framework be trained in end-to-end optimization?
2.	A resource-constrained perspective on OOD generalization is interesting. This paper mainly considers the limited labeled information. What about the limited supervision (such as label noise), limited storage, and limited computation. I think a more comprehensive discussion will further improve this paper and have a wider impact.


**Ethics Review Area:**

["Responsible Research Practice (e.g., IRB, documentation, research ethics)"]

**Limitations:**

The authors provide the discussion about negative societal impact. I suggest the authors make a further discussion about more powerful feature learning technologies with the proposed RA2 framework.



**Strengths And Weaknesses:**

Strengths:
1.	The problem is interesting and the novelty is valuable. Authors consider the limitation of the existing OOD research and theoretically justify their dependence on a large number of labeled data. This paper introduces active learning and addresses OOD generalization via interactive learning. It may inspire some consideration about OOD generalization under limited resources, such as limited label information.
2.	The authors present the generalization analysis and design an active heterogeneity expansion to achieve the ideal maximal invariant predictor. Such a framework is intuitive and theoretically sound.
3.	The experiments include three common distribution shifts: region shift, person shift, and time shift. The experiment results seem sufficient to support the main claim.

Weaknesses:
1.	The invariant learning module in the proposed RA2 is dependent on the previous environmental-based invariant learning methods. The connection between the two sub-modules seems weak. Could the whole framework be trained in end-to-end optimization?
2.	The mask-based feature selection may be weak to more complicated data input, such as high-dimensional images.

---

> ### Author Response · Authors · 2022-08-02
> **Reply to Reviewer nvJG**
>
> Thank you for the advice.
> 1. About the end-to-end optimization. In this paper, we implement the proposed framework in a two-step scheme. The existing framework is not suitable for end-to-end optimization because it relies on an ungeneralizable samples inference module $g$. Nevertheless, the interactive connection of the two sub-modules has been verified by theoretical analysis and ablation experiments.
> 2. About the resource-constrained perspective. It is interesting to consider out-of-distribution generalization under constrained resources. In this paper, we mainly focus on label-efficient methods and present an active model adaptation framework. In the future, we will explore more perspectives to generalization, such as limited storage or computation.

---

### Official Review · Reviewer_WaQo · 2022-07-10

**Rating:** 8
**Confidence:** 5
**Soundness:** 3 good
**Presentation:** 3 good
**Contribution:** 3 good

**Summary:**

This paper studies active learning with Out-of-Distribution (OOD) generalization where target distribution may differ from the source distribution. To address it, the authors present the invariant minimization principle to guide the active learning, interactively learn the invariant relationship and actively query the current un-generalizable samples. Authors formulate it as an Active Heterogeneity Expansion process, and propose the RA2 method, with the corresponding theoretical support. Moreover, the experiment results from the benchmark simulation and real-world tasks clearly verify that the proposed RA2 outperforms SOTA active and OOD methods. Overall, the paper is technically sound and well supported by the corresponding theoretical analysis and empirical evaluation.

**Questions:**

1. About the combination with representation learning
2. About the complexity analysis


**Strengths And Weaknesses:**

Strengths:
1.	The problem of model adaptation under changed distributions is an important problem and has not been well addressed yet. The invariant minimization principle presented by the authors provides a novel view of active learning to address both overall performance and worst-case guarantees. It may motivate researchers to understand active adaptation from causal views and promote robust deployment in real-world applications.
2.	The proposed method RA2 is well-motivated by solid theoretical analysis. Theorem 3.7 presents the generalization conditions under the structural causal model, and indicates the heterogeneity of labeled data is critical to generalization performance. Theorem 3.10 shows the proposed Active Heterogeneity Expansion process could efficiently improve generalization via active interaction.
3.	The experiments seem comprehensive. In both benchmark simulation and real-world tasks, RA2 outperforms the SOTA active adaptation and OOD methods. The effectiveness of the proposal has been clearly verified.
4.	The paper is well organized and easy to follow. The related active adaptation and OOD literature have been comprehensively reviewed and discussed. The presentation of RA2 is generally clear.

Weaknesses:
1.	The proposed invariant learning module (Sec. 4.2) focuses on mask selection and raw-level features. The former framework (Line 167-174, Sec. 4) seems not limited to raw-level selection. There is also a discussion about representation learning in the appendix. I think the feature selection, presented in Section 4.2, could be further improved, with consideration of representation learning.
2.	There are two interactive modules in the proposed RA2. Compared to previous active adaptation methods, which are designed on a specific metric, it introduces more computation processes. How about the complexity compared with previous methods?
3.	Illustration: The text in Figures 2, and 4 is too small. It should be adjusted to the same size as Figures 1, and 3.

---

> ### Author Response · Authors · 2022-08-02
> **Reply to Reviewer WaQo**
>
> Thank you for the advice.
> 1. The proposed framework RA2 is not limited to raw-level feature selection. In this paper, we mainly introduce and evaluate our framework based on raw-level features. Through the experiments on C-MNIST (experiments reported in the appendix), the feasibility of combination with representation learning has been demonstrated. We would like to explore more about representation learning in future work.
> 2. About the complexity analysis. Compared with the regular active strategies, we introduce two sub-modules: invariance optimization and distribution inference. The theoretical convergence rate of invariant learning is still a challenge in the invariant learning community. Empirically, it could converge with small constant iterations. The latter could achieve a closed-form solution in $O(d^3+d^2 N_T+dN_S)$. Considering annotation cost is more critical than computation cost in active learning, the overhead of our polynomial cost is acceptable. Empirically, we could derive our solution in 5 seconds (device: RTX 3090) for 380000 instances.

---

### Official Review · Reviewer_8KWW · 2022-07-10

**Rating:** 5
**Confidence:** 3
**Soundness:** 2 fair
**Presentation:** 3 good
**Contribution:** 3 good

**Summary:**

The paper presents a method for active adaptation of a model trained on source data, when target data from changed distribution is available. It builds on the invariance minimization principle and extends the so called Maximal Invariant Predictor (MIP) for determining where to sample points from the unlabeled target data. The authors show that increasing the heterogeneity of training data improves generalization, and hence queries from samples where the current invariant model does not hold. Representative points are sampled from these using coreset. The model is updated by using a new invariance predictor obtained from the updated data.

The experiments include comparison with relevant baselines. Results are shown on synthetic data (along with the case where the target is an imbalanced mixture), and on three real world datasets.

**Questions:**

The active adaptation baselines included in the paper show results on DIGITS, OFFICE, etc datasets -- a comparison on those datasets is imperative. Also, the current results are using a fixed budget size. It would be helpful to add the effect of different budget size on the method, and the time complexity of querying samples (w.r.t size/classes/budget of target data).

The method requires a knowledge of mixture proportion in the unlabeled target data. Does this limit the quality of the approach in different scenarios?
A more thorough contextualization with related active adaptation methods would be helpful for readers.

**Limitations:**

The limitations of the work have not been described. Potential societal impact of the work has been discussed.

**Strengths And Weaknesses:**

The paper provides an interesting extension of invariant minimization principle for active adaptation. The overall idea of the paper makes sense. The results on synthetic and small real data shows improvements in overall and worst-case error. Feature importance results and ablation studies further shows efficacy of the method on these datasets.

Comparisons on standard datasets used in other active adaptation methods are missing. Discussion/experiments to help understand the effect of different practical choices would be helpful.

---

> ### Author Response · Authors · 2022-08-02
> **Reply to Reviewer 8KWW**
>
> Thank you for the advice.
> 1. About the experiments. Except for image data, this paper also studies tabular data with widely real-world applications. These datasets are widely employed in the related out-of-distribution generalization research [1, 2, 3]. Note that our experiments have included different distribution shift types: region, person group, and time. As a result, the presented method saves more than 50% samples to achieve competitive performance compared to the baseline (Figures 4 d and e), showing it has efficiently extracted the invariance and promoted active adaptation. For image data, we also reported the results on the benchmark C-MNIST in the appendix (Section C.6, pages 7-8). It has clearly demonstrated the effectiveness of our framework with representation learning. In addition, we had reported the active learning results on real-world data with varying budgets in Figure 4 (page 8 in the main paper).
>
> 2. About the knowledge of mixture proportion. For the knowledge of mixture proportion, we have provided its estimation method (line 202-204, page 5) and implemented it in the experiments. All of our results are reported by the estimated mixture proportion. Notice that the mixture proportion estimation is well studied and not our main contribution. Thus, we omit the details in this paper and refer readers to the corresponding implementation [4].
>
> [1] Heterogeneous risk minimization. ICML 2021
> [2] Environment Inference for Invariant Learning. ICML 2021
> [3] Kernelized heterogeneous risk minimization. NeurIPS 2021
> [4] Mixture proportion estimation via kernel embeddings of distributions. ICML 2016

---

> > ### Comment · Reviewer_8KWW · 2022-08-09
> > **Comments on the rebuttal**
> >
> > Thanks to authors for their reply.
> >
> > As the paper tackles active adaptation, and all three active adaptation baselines considered in the paper [37, 28, 6] show results on datasets like DIGITS and OFFICE, I still believe it would make a stronger paper with comparisons on these datasets. I acknowledge that the results on real world tabular data, and distribution shifts on them, validates the novel method to an extent.
> >
> > Thank you for clarification on the mixture proportion.

---

### Official Review · Reviewer_1tYQ · 2022-07-11

**Rating:** 5
**Confidence:** 5
**Soundness:** 3 good
**Presentation:** 2 fair
**Contribution:** 3 good

**Summary:**

This paper presents the invariance minimization principle, to address to handle distribution shift at a relatively small labeling cost. For this purpose, this paper designs an interactive model adaptation framework including active sample selection and invariant relationship learning. The authors demonstrate the effectiveness of the method both theoretically and experimentally.

**Questions:**

What is the physical meaning of the semantic variable and intervention variable and how these two variables are reflected in the experiment？

**Limitations:**

The author does not mention the limitations of the work.

**Strengths And Weaknesses:**

On one hand, the strengths include:
1.	The paper is well-written.
2. The theoretical work of this paper is sufficient, which improves the value of the paper.
3.	The paper has several novelties: i)
 i) This paper proposes the invariance minimization principle for active model adaptation; ii) For this purpose, The authors present an interactive framework Robust Active model Adaptation (RA2); iii) This paper demonstrates the effectiveness in an extensive empirical study and theoretical analysis.

On the other hand, for me, the main weakness mainly lies in that the paper is not easy to follow. I understand that the authors have put necessary proofs and details into the supplementary material, but still Sections 3 are not easily understandable. The author explains some symbols unclearly, e.g., what is the physical meaning of the semantic variable and intervention variable and how these two variables are reflected in the experiment？ In addition, the use of formulas is not uniform, e.g., why does the formula \Phi(S(X^{e})) use observation S in line 118, but not in line 120?

---

> ### Author Response · Authors · 2022-08-02
> **Reply to Reviewer 1tYQ**
>
> Thank you for the advice.
> Semantic variable and intervention variable are terms in causality, that is, the semantic variable causals the instance label $Y$: $Y=f(Z^s)$ in our Assumption 3.3. More specifically, the intervention variable is the latent variable in $X$ that is irrelevant to the task label $Y$. Learning model directly on $(X, Y)$ may overfit the spurious correlation between intervention variables and task label. Under distribution shift, the relation between the intervention and task label will shift, but the relation between semantic variables and task label holds stable. More importantly, for out-of-distribution generalization, our goal is to identify the semantic variable (cause) to predict $Y$ and reduce the overfitting of intervention variables. In our experiments on real-world datasets, we validate the robustness on three types of intervention variables (distribution shifts): region, person, and time. For example, in income prediction, the features “sex” and “race” are the intervention variables. The features “work class”, “education”, “hours-per-week” … are the semantic variables that cause the income $Y$. Our learning model is not dependent on the intervention variables, and obtains a _robust_ predictor across different person groups.

---

### Meta-Review · Area_Chair_oTGX · 2022-08-24

**Recommendation:** Accept
**Confidence:** Certain

**Metareview:**

The reviews and the discussions converged on the consensus that the paper contains novel ideas and is theoretically solid.

However, a discrepancy between the scores remains after the discussions due to different opinions on the experimental part, especially the lack of comparison with standard adaptation baselines in the computer vision community. I read the manuscript, and I agree with reviewer WaQo and nvJG that the experiments are sufficient to support the claims, considering that the proposed method does not naturally apply to image data.

That being said, I kindly ask the authors to take into account reviewers' comments while preparing the camera-ready version.

**Award:**

No

---

### Decision · Program_Chairs · 2022-09-14

Accept